# Evaluation of the Bio-Stimulating Activity of Lake Algae Extracts on Edible Cacti *Mammillaria prolifera* and *Mammillaria glassii*

**DOI:** 10.3390/plants11243586

**Published:** 2022-12-19

**Authors:** Domenico Prisa, Damiano Spagnuolo

**Affiliations:** 1CREA Research Centre for Vegetable and Ornamental Crops, Via Dei Fiori 8, 51012 Pescia, Italy; 2Department of Chemical, Biological, Pharmaceutical, and Environmental Sciences, University of Messina, Salita Sperone 31, 98166 Messina, Italy

**Keywords:** sustainable agriculture, bio-fertilisers, organic farming, seaweed extracts, edible cactus, ornamental plants

## Abstract

The research aimed to test different seaweed extracts derived from three macroalgae representatives, namely, *Rhodophyta, Chlorophyta* and *Phaeophyceae*, as a bio-fertiliser for the growth of *Mammillaria prolifera* and *Mammillaria glassii* and the production of edible fruits. The experiments started in September 2021 and were conducted in the greenhouses of CREA-OF in Pescia (PT). Three different algae, namely, *Hypnea cornuta* (*Rhodophyta*), *Ulva ohnoi* (*Chlorophyta*), collected from the brackish lake Ganzirri, in Messina, and *Sargassum muticum* (*Phaeophyceae*) from Venice lagoon, were tested. The experimental trial showed a significant improvement in the agronomic parameters analysed for the growth and production of cactus plants and fruits treated with the selected algae. A significant increase was found in the sugar, vitamin A, vitamin C and vitamin E content of the fruits of treated plants. In particular, the thesis with *Ulva ohnoi* was the best for plant growth and fruit production with a higher sugar and vitamin content. This experiment confirms the algae’s ability to stimulate soil microflora and microfauna, promoting nutrient uptake, participating in organic matter mineralisation processes and significantly influencing the nutraceutical compounds in the fruits.

## 1. Introduction

Anthropic processes act as one of the prominent effects accentuated by climate change. Consequently, drought events are increasing, causing a relevant loss in crop yield and water availability for cultivation [1,2]. Considering this, improving resistance ability and agricultural production is vital, and is a challenging task, but is needed to meet the growing food demand due to population growth [3]. Optimal mineral nutrition increases plant resilience to different stresses and enhances products’ quality. Nevertheless, chemical fertilisation causes high economic and ecological costs [4]. Fertilisers from macroalgae products that respond to the requirements for agriculture could be used in biological agriculture as they are not synthetic, and at the same time, they can provide greater yields and mitigate the effects of climate change, stimulating plant nutrition processes [5,6,7]. Additionally, macroalgae fertiliser provides a wide range of plant growth regulators, thus turning this fertiliser into a more noble biostimulant [8,9,10]. Therefore, adopting macroalgae as a bio-fertiliser is a good solution for sustainable agriculture because it combines an economical source to obtain fertiliser with the removal of some macroalgae biomass, which may occur especially in eutrophic environments [11,12]. Macroalgae can produce immense biomass in nutrient-rich environments, such as estuaries, natural and artificial lakes, and, more generally, in transitional environments that often have a substantial anthropogenic impact, which is often dredged as a waste so as not to affect human activities. Previous experiments using algae on plants such as maize and rice resulted in increased growth in terms of dry matter and the C and N content of the soil [13,14,15]. In addition, an increase in shoot height in the treated maize theses and an increase in root and shoot development in rice were also observed. Tests on *Amaranthus* spp. showed an increase in plant height and biomass produced in the theses treated with algae [16,17]. Further trials on *Portulaca grandiflora*, *Aloe Barbadensis Miller* and *Lobivia* spp. showed a significant effect on increased vegetative and root development of the plants and on the mineral and sugar content in stems and leaves [18,19,20]. Algae extracts have been used for a long time because they can improve plant growth and increase resistance to various biotic and abiotic stresses. In particular, algae contain considerable amounts of indispensable mineral elements such as calcium, phosphorus, iron and potassium; various compounds with hormonal and stimulating activity are also present in algae [21,22]. In addition, algae contain colloid substances used for various purposes, e.g., for their proven stimulating activity [23,24]. Agriculture consumes a considerable amount of nutrients, especially nitrogen and phosphorus, to meet the food demands of an actively growing global population. However, nutrients to sustain the agricultural system are supplied by unsustainable processes, and most fertilisers introduced into agriculture are dispersed into the surrounding ecosystem, which, when dispersed into the soil, bind with the organic matter complexly, making plants unable to assimilate them, thus burdening the environmental impact of the agricultural system [25]. Therefore, developing sustainable alternatives for nutrient production to sustain the current agricultural production chain is a priority. Among the alternatives, the high content of micro- and macronutrients makes microalgae biomass a promising source of bio-fertiliser. From different algal species, bio-fertilisers with stimulating properties are generally produced individually or in combination with bacteria and mycorrhizae. Microalgae and cyanobacteria products are generally developed by means of photobioreactors, such as *Arthrospira* spp., *Dunaliella* spp., *Anabaena* spp., *Phaeodactylum* spp., *Pleurochrysis* spp., *Chlorella* spp. and *Nannochloropsis* [26,27,28]. These products, once obtained, can be applied to plants in fertigation or combined with fertilisers in integrated agriculture protoclimates. Stimulant products based on *Ascophyllum nodosum* are used today as stimulants for increasing plant protection from biotic and abiotic stresses in ornamentals [29]. Various microalgae species have been studied for their application as bio-fertilisers, with soil-stabilising effects, increased nutrient content, and increased water retention capacity. However, the mechanism responsible for bio-fertilisation has not yet been fully elucidated [30]. Microalgae have a high nutrient content, which must be accessible to plants, so the microbiome of the rhizosphere degrade the biomass of the microalgae to release constituent nutrients or undergo natural degradation to allow a sustained release of nutrients. Alternatively, microalgae biomass could actively interact with plants and, in the case of nitrogen-fixing cyanobacteria, induce the release of bioavailable forms of nitrogen in exchange for carbon compounds from the plant [31]. In the latter case, the microalgae would also have to interact with the rhizosphere microbiome actively, and, therefore, compatibility and survival are not guaranteed. Combined with the possibility of providing energy resources, microalgae biomass has a broader effect on plant growth by synthesising psychostimulant molecules such as hormones [32,33]. Phytohormones found in extracts of microalgae biomass include auxins, gibberellin-like molecules and abscisic acid, affecting the growth and plant development due to stimulatory activity on various metabolic processes such as photosynthesis, respiration, nucleic acid synthesis and nutrient assimilation [34].

The research aimed to test different seaweed extracts, derived from three macroalgae representatives of the *Rhodophyta*, *Chlorophyta* and *Phaeophyceae*, as a bio-fertiliser for the growth of *Mammillaria prolifera* and *Mammillaria glassii* and the production of edible fruits, and to evaluate the interactions between algae and the soil microbiome (Figure 1).

## 2. Results

The experimental trial at the CREA-OF greenhouses in Pescia showed a significant improvement in the agronomic parameters analysed on the growth and production of *Mammillaria prolifera* and *Mammillaria glassii* fruits treated with selected algae from Lake Ganzirri (ME) and the Venice Lagoon. Treatments with selected algae from these lakes resulted in an increase in the vegetative and root development of the plants, with an increase in height, circumference and number of spines and flowers in the epigeal part and root length in the hypogeal part of these plants. Significant effects were also evident in the increase in the microbial biomass of the algae-treated theses. Finally, a significant increase in sugar, vitamin A, vitamin C and vitamin E content was found in the fruits of the plants treated with the stimulating algae. In particular, the thesis with *Ulva ohnoi* was the best of all, both in terms of plant growth and the sugar and vitamin content of the fruits. The results show the differences in plant growth, fruit production and sugar and vitamin content obtained with the treatments of (AG) Ecklonia maxima, (HC) Hypnea cornuta, (UO) Ulva ohnoi and (SM) Sargassum muticum compared with the fertilised control (CTRL).

Mammillaria prolifera plant height was significantly increased by UO at 9.38 cm, SM at 8.25 cm, AG at 8.23 cm, HC at 8.14 cm and finally (control) at 7.46 cm (as shown in Table 1). With 4.4 new suckers, UO was the best thesis, followed by SM with 3.4, HC with 3.2, AG with 2.6 and CTRL with 1.6. Significant development of the vegetative part can be seen in UO, followed by AG and HC at 34.97 and 34.86 g, respectively, SM at 34.37 g, and finally, CTRL at 32.89 g (Figure 2). Figure 3 shows the same trend in root weight, with the UO root weight being 25.75 g, the SM and the HC root weights being 23.14 g and the AG root weight being 22.93 g. In terms of plant circumference, UO was also the best thesis with 6.63 cm, followed by SM at 5.86 cm, HC at 5.80 cm, AG at 5.29 m and CTRL at 4.86 cm. Ulva ohnoi was also the best thesis for flower production, 22.6, compared to 15.4 (AG), 14.8 (HC), 14.4 (S, M) and 13.6 (CTRL). In flower duration, UO was the best thesis with 4.6 days, followed by AG and HC with 3.6 days, (M with 3.4 days, and finally, the control with 2.4 days. No major differences in the development of the spines are evident between the various experimental theses.

There were no significant changes in the pH of the substrate (Table 2) of Mammillaria prolifera. As a result of the test, algae-treated substrates demonstrated a higher microbial presence with 1.29 × 10^3^ cfu/g, followed by HC with 1.27 × 10^3^ cfu/g, SM with 1.26 × 10^3^ cfu/g, AG with 1.23 × 10^3^ cfu/g and finally the control with 1.25 × 10^2^ cfu/g. The thesis UO had the highest fruit production with 14, compared to HC with 7.8, AG and SM with 7.4 and the control with 5.8. There was also a similar trend in fruit weight, with (UO) having 3.72 g (the best fruit weight), HC having 3.33 g and SM having 3.29 g, respectively, whereas AG having 3.17 g and CTRL having 2.50 g. In terms of sugar content, UO and SM topped the list with 4.85 g and 4.82 g, respectively, followed by HC with 4.76 g, AG with 4.68 g and CTRL with 4.57 g. Vitamin A content in the SM thesis was significantly lower than in the others, while vitamin C and vitamin E contents in the UO and SM thesis were the highest.

The plant height of UO of Mammillaria glassii was significantly increased at 6.48 cm (Table 3), followed by SM at 6.41 cm, HC at 6.31 cm, AG at 6.29 cm and finally, CTRL at 5.38 cm. There were 3.4 new suckers in UO, AG and HC, followed by SM with 1.6 and the control with 1.2. As shown in Figure 4, UO had a better weight for the vegetative part at 29.23 g, followed by HC at 28.27 g, AG at 27.87 g, SM at 27.48 g and CTRL at 26.46 g. Root weight follows the same trend with UO weighing 20.13 g, AG and HC weighing 18.87 g and 18.86 g, respectively, and, finally, SM depicting 17.74 g, as well as the untreated control of 17.68 g. Plant circumference was also highest for UO at 6.25 cm, followed by HC and SM at 5.94 cm and 5.83 cm, respectively, AG at 5.27 cm and CTRL at 4.63 cm. In Mammillaria glassii, Ulva ohnoi had the highest flower production, 15.0, compared with 12.8 for AG and HC, 12.4 for SM and 8.6 for CTRL. The flower duration in UO was highest at 4.4 days, followed by AG and HC at 3.8 days, 3.6 days and, finally, 2.8 days for SM and CTRL. Differences in the growth of thorns were evident, with UO showing the best results having 124.2 spines of 3.06 cm in length, respectively.

Mammillaria glassii substrate pH hovered around 6.4 (Table 4), but no significant difference was observed. One of the most interesting aspects of this test was the microbial colonisation of the algae-treated substrates; in particular, UO was the thesis with the highest microbial presence of 1.64 × 10^3^ cfu/g, followed by HC and SM with 1.32 × 10^3^, AG with 1.26 × 10^3^ and CTRL with 1.25 × 10^2^. As compared to all of the other experimental theses, UO produced the best fruit, with 11.4. According to fruit weight, UO produced the best weight at 4.07 g, followed by SM at 3.84 g, AG at 3.72 g, HC at 3.29 g and CTRL at 2.43 g. With 4.55 g of sugar and 7.30 mg of vitamin, UO had the highest levels of sugar and vitamin content. Vitamin C content in all theses with stimulating algae was higher than that in control, while vitamin E content in UO was the highest (0.06 mg).

## 3. Discussion

It seems that the term biostimulant was coined to describe those substances that promote plant growth without being nutrients, soil conditioners or pesticides. Interestingly, the first discussion of biogenic stimulants is attributed to a Russian named Filatov and dates back to the 1930s [35,36]. This definition refers to specific biological material derived from various organisms, including plants, which have been exposed to stressors and could influence metabolic and energetic processes in humans, animals and plants [37,38,39]. Depending on the geographical region and species, climate change can have positive and negative effects on crops and can impact plant growth, fruit development, flower intensity and structure. Despite being cultivated and well-watered, the impact on plants is clearly evident and leads them to become smaller, shorter and less drought-resistant. In addition, the amount of CO_2_ absorbed through photosynthesis is reduced, resulting in a significant reduction in plant productivity. Climate change can also lead to the development of serious plant diseases caused by fungi and insects that usually live in different areas [40]. Numerous studies have been conducted to identify the functional molecules contained in algae extracts. According to Prisa [41] and Mulberry [42], the use of algae can stimulate plant growth by influencing tissue mineral content, flower production and plant quality. Among the bioactive substances that seem to be responsible for this growth, hormones are certainly the most likely growth stimulators; although, they cannot always be determined analytically because they are often contained in concentrations below the sensitivity threshold of instruments [43,44]. The hormones most commonly detected are cytokinins, auxins, gibberellins and abscisic acid, and, as in other experiments, they also influenced plant height and girth as well as root development in this trial. Traces of ethylene precursors, which promote flowering and fruit ripening in sensitive species, were found in some algae extracts; this effect was also evident in flowers and fruits of Mammillaria. In the past, the positive effects of algae and their extracts on crop productivity have been attributed to the supply of organic matter and, thus, to the improvement in physical, chemical and biological soil fertility [45,46]. However, these effects cannot explain the beneficial action of algae extracts administered to crops in liquid form at extremely low dosages [47]. Recent experimental evidence has shown that liquid algae extracts at low doses manifest positive effects on plant growth, health and crop yield through an action that cannot be attributed to nutrient supply [48,49]. In particular, the biological action of algae extracts has been shown to be due to the presence of carbohydrates, amino acids, vitamins, traces of hormones and hormone-like substances [50]. Studies by Faheed and El Fattah (2008), for example, highlighted the effects of the algae *Chlorella vulgaris* on *Lactuca sativa* L., noting that the presence of the algae accelerated the seed germination process and increased the chlorophyll a and b and carotenoid content [51]. Agwa et al. (2017), using *C. vulgaris* on *Hibiscus esculentus* as a replacement for chemical fertilisers, showed improved plant growth and biomass production [52]. The algae have a high content of primary metabolites, carbohydrates and lipids (55–70% of fresh weight) [53], which certainly interact with plant metabolism, as was also confirmed in this trial. Generally speaking, in *Chlorella* spp., *Chlamydomonas* spp., *Dunaliella* spp. and *Spirulina* spp., carbohydrates can make up 46% of the dry extract, while proteins account for 18–46% of the dry extract. The presence of certain amino acids, such as tryptophan and arginine, in algae extracts, can cause a significant increase in crop growth and yield, as these amino acids are the metabolic precursors of phytohormones. Algae also exhibit an indirect bio-stimulating effect, attributed to their ability to increase and modify the microbial component of the soil, with concomitant effects on nutrient mineralisation as well [54,55,56]. In experiments on Mammillaria, algae-treated theses showed a significant increase in microbial biomass. Several studies have indeed demonstrated an increase in the soil microbiota following the inoculation of algae, whose prolific production of extracellular polymeric substances served as a carbon source for plants and bacteria in the rhizosphere [57]. The trial emphasised the bio-stimulating capacities of new algae found in Lake Ganzirri and the Venice Lagoon, which had previously not been tested on the growth and productivity of cactus plants, resulting in a significant increase in agronomic parameters, but also in the nutraceutical parameters of the fruits. The latter aspect was certainly influenced by the bioactive compounds of the algae extracts, which stimulated the physiological and biochemical processes of the growing plants [58]. The Mammillaria fruits also showed a better yield than the control and an increase in antioxidant capacity and vitamin C content [59]. Data on *Cucumis sativus* L. also confirmed the same where significant improvements in plant metabolism occurred following treatments with algae such as *Macrocystis pyrifera*, *Bryothamnion triquetrum*, *Ascophyllum nodosum*, *Grammatophora* spp. and *Macrocystis integrifolia*, or on Annurca apple fruits, which showed an increase in total soluble solids (TSS) content, total acidity (TA), pH, flesh consistency and red colouration of the epicarp at harvest stage. The treated fruits also showed a significantly higher content of total polyphenols in the pulp and a higher concentration of xyloglucoside and floridzin and an increase in flavonols during cold storage [60]. In addition, algae extracts applied to substrates improve the soil microflora by increasing its biological fertility, thus creating a favourable environment for root growth, which increases the exploration capacity of the soil and indirectly influences the bioavailability of nutrients [61]. Indeed, many scientific works have shown how algae extracts can promote crop growth and increase the yield of herbaceous and tree crops and the quality of edible products [34]. The main responses of herbaceous and tree species include increased seed germination rate, root growth, leaf quality, vigour and tolerance to abiotic stresses. Some extracts increase flowering and fruiting and also improve product quality [61].

## 4. Materials and Methods

Three different macroalgae were sampled and used to prepare the fertiliser: *Hypnea cornuta* (*Rhodophyta*) and *Ulva ohnoi* (*Chlorophyta*) collected from the brackish lake Ganzirri, in Messina, Italy (38°15′28” N–15°36′37” E) and *Sargassum muticum* (*Phaeophyceae*) from Venice lagoon, Italy (45°25′42.6” N–12°19′50.7” E). After sampling, biomasses were transported in chilled condition after cleaning by washing in fresh water and drying in an oven at 40 °C for 48 h until extraction. For the preparation of Liquid Seaweed Fertilizer (LSF), dried macroalgae were prepared with the modified protocol from Rama (1990) [1]; the process is described in detail in Spagnuolo and Prisa (2021) [2]. In summary, each solution was prepared with 500 mL of distilled water and 25 g of dried seaweed (ratio 1:20 DW/V) at 80 °C for three hours. The residue biomass was removed using a cotton cloth, and the liquid solution gave an SLF that was used in different concentrations in irrigation water on *Mammillaria prolifera* and *Mammillaria glassii*. The plants used were chosen because they produce fruit of edible and medicinal interest and could also be appreciated in the future in view of climate change and the low maintenance they require. To preserve the fertiliser, 1 g/l of citric acid is added and kept at 4 °C until use. The experiments, initiated in September 2021, were conducted in the greenhouses of CREA-OF in Pescia (PT), Tuscany, Italy (43°54′ N 10°41′ E), on *M. prolifera* and *M. glassii*. The plants were grown in ø 12 cm pots, 30 plants per thesis, divided into three replicas of 10 plants each. Plants were fertilised with a controlled-release fertiliser (3 kg m3 Osmocote Pro^®^, 9–12 months) added to the growing medium of the plants. The experimental groups were as follows:Group control (CTRL) (peat 30% + pumice 70%), irrigated with water and substrate previously fertilised;Group with algae (AG) (peat 30% + pumice 70%) irrigated with water and substrate previously fertilised, dilution 1:1000 once a week (Kelpak biostimulant, *Ecklonia maxima*, Kelp products International);Group with *Hypnea cornuta* (HC) (peat 30% + pumice 70%) irrigated with water and substrate previously fertilised, dilution 1:1000 once a week;Group with *Ulva ohnoi* (UO) (peat 30% + pumice 70%) irrigated with water and substrate previously fertilised, dilution 1:1000 once a week;Group with *Sargassum muticum* (SM) (peat 30% + pumice 70%) irrigated with water and substrate previously fertilised, dilution 1:1000 once a week.

The concentration and frequency of use in the trial were based on the use of commercial algae, already applied in agriculture. In anticipation of the possible use of these algae and their possible marketing, an attempt was made to compare the protocols and concentrations of use with already tried and tested products. The plants were watered four times a week and cultivated for nine months using an automatic irrigation technology system. On 10 May 2022, plant height and circumference, suckers number, number and length of thorns, vegetative weight, root weight, number of flowers, flowers’ life, number and weight of fruits, substrate microbial count, pH, sugar content (g), vitamin C (mg), vitamin A (mg) and vitamin E (mg) concentration were analysed according to Al-Mhanna et al. (2018) [60]. Direct determination of the total microbial count by microscopy of the cells contained in a known volume of sample was performed using counting chambers (Thoma chamber). The surface of the slide is etched with a grid of squares, with the area of each square known. Determination of viable microbial load after serial decimal dilutions, spatula seeding (1 mL) and plate counting after incubation was performed [61].

### Statistics

A randomised block design was used in the experiment and the data obtained were analysed according to the one-way ANOVA to assess whether significant differences existed between the various experimental theses. Then, mean values were separated by LSD multiple-range test (*p* = 0.05). Graphics and statistics were supported by the programs Costas (version 6.451) and Excel (Office 2010).

## 5. Conclusions

Algae can colonise almost any type of habitat, though most of them live in seas, oceans and fresh waters. However, they can colonise other environments, including deserts, volcanic waters, very acidic or frozen soils, rocks, plants and artificial substrates as well. In the terrestrial environment, algae, especially cyanobacteria, contribute to pools of soil organic matter, either directly or through the secretion of exopolysaccharides and the production of humus-like substances. In addition, they can stimulate the growth of other microflora and microfauna, promote nutrient uptake and participate in organic matter mineralisation processes.

## Figures and Tables

**Figure 1 plants-11-03586-f001:**
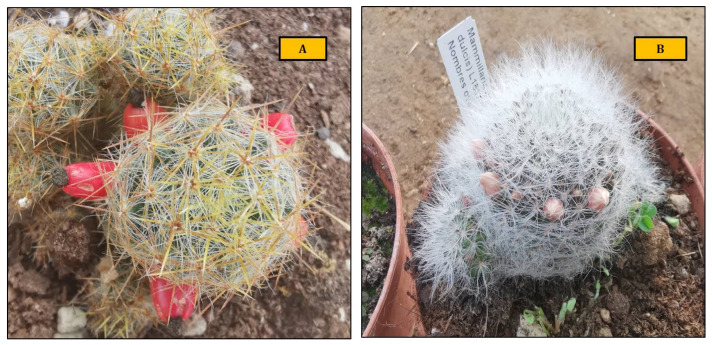
A view of *Mammillaria prolifera* (**A**) and *Mammillaria glassii* (**B**) plants in CREA-OF greenhouses.

**Figure 2 plants-11-03586-f002:**
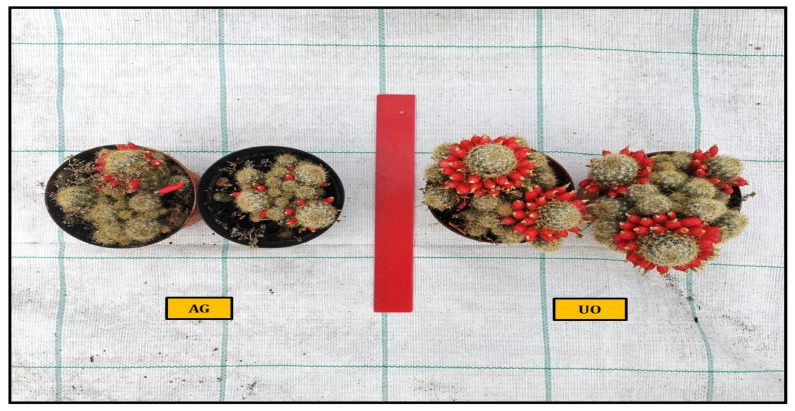
Comparison of vegetative growth and fruit production of *Ulva ohnoi* (UO) and *Ecklonia maxima* (AG) in *Mammillaria prolifera*.

**Figure 3 plants-11-03586-f003:**
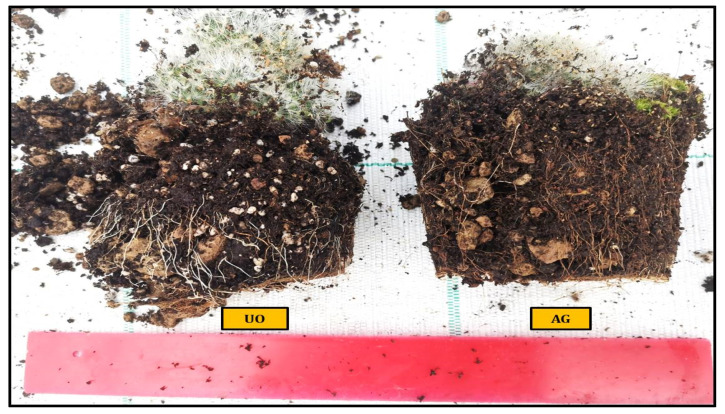
Comparison of root growth in *Mammillaria prolifera* plants treated with *Ulva ohnoi* (UO) and *Ecklonia maxima* (AG).

**Figure 4 plants-11-03586-f004:**
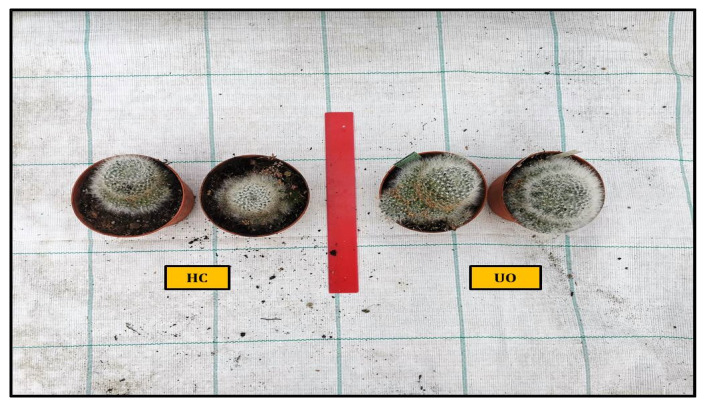
Comparison of the vegetative growth of *Ulva ohnoi* (UO) and *Hypnea cornuta* (HC) in *Mammillaria glassii*.

**Table 1 plants-11-03586-t001:** Evaluation of liquid seaweed fertiliser on agronomic characters of *Mammillaria prolifera*.

Groups	PH (cm)	SN (n°)	VW (g)	RW (g)	PC (cm)	FN (n°)	FL (days)	NT (n°)	LT (mm)
CTRL	7.46 c	1.6 d	32.89 d	21.55 c	4.86 d	13.6 c	2.6 c	95.0 a	2.58 a
AG	8.23 b	2.6 c	34.97 b	22.93 b	5.29 c	15.4 b	3.6 b	96.4 a	3.28 a
HC	8.14 b	3.2 bc	34.86 b	23.14 b	5.80 b	14.8 b	3.6 b	95.6 a	3.36 a
UO	9.38 a	4.4 a	37.72 a	25.75 a	6.63 a	22.6 a	4.6 a	95.0 a	10.3 a
SM	8.25 b	3.4 b	34.37 c	23.14 b	5.86 b	14.4 bc	3.4 b	96.0 a	2.74 a
ANOVA	***	***	***	***	***	***	***	*ns*	*ns*

Significance according to Tukey’s analysis (HSD) multiple-range test (*p* = 0.05). Legend: (CTRL): control; (AG): *Ecklonia maxima*; (HC): *Hypnea cornuta*; (UO): *Ulva ohnoi*; (SM): *Sargassum muticum*; PH: plant height; SN: suckers number; VW: vegetative weight; RW: roots weight; PC: plants circumference; FN: flowers number; FL: flowers life; NT: Thorns number; LT: thorns length. Different letters in the same column mean significant differences between varieties.

**Table 2 plants-11-03586-t002:** Evaluation of microbial biomass, production, sugar and vitamin content in *Mammillaria prolifera fruits*.

Groups	PH	MC (cfu/g)	FN (n°)	FW (g)	SC (g)	Vit. A (mg)	Vit. C (mg)	Vit. E (mg)
CTRL	6.48 a	1.25 × 10^2^ d	5.8 c	2.50 d	4.57 d	7.36 a	8.45 c	0.024 c
AG	6.48 a	1.23 × 10^3^ c	7.4 b	3.17 c	4.68 c	7.24 a	8.62 b	0.026 c
HC	6.48 a	1.27 × 10^3^ b	7.8 b	3.33 b	4.76 b	7.26 a	8.72 b	0.054 ab
UO	6.44 a	1.29 × 10^3^ a	14.0 a	3.72 a	4.85 a	7.32 a	8.95 a	0.064 a
SM	6.44 a	1.26 × 10^3^ b	7.4 b	3.29 b	4.82 a	6.98 b	8.86 a	0.044 b
ANOVA	*ns*	***	***	***	***	***	***	***

Significance according to Tukey’s analysis (HSD) multiple-range test (*p* = 0.05). Legend: (CTRL): control; (AG): *Ecklonia maxima*; (HC): *Hypnea cornuta*; (UO): *Ulva ohnoi*; (SM): *Sargassum muticum*; PH: acidity or basicity of the substrate; MC: microbial count; FN: fruit number; FW: fruit weight; SC: sugar content. Different letters in the same column mean significant differences between varieties.

**Table 3 plants-11-03586-t003:** Evaluation of liquid seaweed fertiliser on agronomic characters of *Mammillaria glassii*.

Groups	PH (cm)	SN (n°)	VW (g)	RW (g)	PC (cm)	FN (n°)	FL (days)	NT (n°)	LT (mm)
CTRL	5.38 d	1.2 c	26.46 e	17.68 c	4.63 d	8.6 c	2.8 c	97.0 c	1.44 d
AG	6.29 c	2.2 b	27.87 c	18.87 b	5.27 c	12.8 b	3.8 ab	111.8 b	2.22 c
HC	6.31 c	2.2 b	28.27 b	18.86 b	5.94 b	12.8 b	3.8 ab	100.4 c	2.42 b
UO	6.48 a	3.4 a	29.23 a	20.13 a	6.25 a	15.0 a	4.4 a	124.2 a	3.06 a
SM	6.41 b	1.6 bc	27.48 d	17.74 c	5.83 b	12.4 b	3.6 b	99.0 c	2.16 c
ANOVA	***	***	***	***	***	***	**	***	***

Significance according to Tukey’s analysis (HSD) multiple-range test (*p* = 0.05). Legend: (CTRL): control; (AG): *Ecklonia maxima*; (HC): *Hypnea cornuta*; (UO): *Ulva ohnoi*; (SM): *Sargassum muticum*; PH: plant height; SN: suckers number; VW: vegetative weight; RW: roots weight; PC: plants circumference; FN: flowers number; FL: flowers life; NT: Thorns number; LT: Thorns length. Different letters in the same column mean significant differences between varieties.

**Table 4 plants-11-03586-t004:** Evaluation of microbial biomass, production, sugar, and vitamin content in *Mammillaria glassii* fruits.

Groups	PH	MC (cfu/g)	FN (n°)	FW (g)	SC (g)	Vit. A (mg)	Vit. C (mg)	Vit. E (mg)
CTRL	6.46 a	1.25 × 10^2^ d	5.6 c	2.43 e	3.70 e	6.14 d	7.36 b	0.03 bc
AG	6.42 a	1.26 × 10^3^ c	8.2 b	3.72 c	4.26 c	7.11 b	7.90 a	0.02 c
HC	6.44 a	1.32 × 10^3^ b	7.6 b	3.29 d	4.44 b	7.13 b	7.93 a	0.03 bc
UO	6.46 a	1.64 × 10^3^ a	11.4 a	4.07 a	4.55 a	7.30 a	8.05 a	0.06 a
SM	6.48 a	1.32 × 10^3^ b	7.4 b	3.84 b	4.18 d	6.98 c	7.95 a	0.04 b
ANOVA	*ns*	***	***	***	***	***	***	***

Significance according to Tukey’s analysis (HSD) multiple-range test (*p* = 0.05). Legend: (CTRL): control; (AG): *Ecklonia maxima*; (HC): *Hypnea cornuta*; (UO): *Ulva ohnoi*; (SM): *Sargassum muticum*; PH: acidity or basicity of the substrate; MC: microbial count; FN: fruit number; FW: fruit weight; SC: sugar content. Different letters in the same column mean significant differences between varieties.

## Data Availability

All data, tables and figures in this manuscript are original.

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
