# Peer review of "Evaluation of the Bio-Stimulating Activity of Lake Algae Extracts on Edible Cacti Mammillaria prolifera and Mammillaria glassii"

_plants, 2022, doi:10.3390/plants11243586_

Round 1

Reviewer 1 Report

This is a great, well-conducted investigation comparing the effect of three different seaweed extracts (from three different algae) on the growth and fruit production of two cactus plants, Mammillaria prolifera and Mammillaria glassii. The experimental trial showed a significant improvement in the some parameters analyzed on the growth and production of fruits treated with selected algae. Although the research is interesting and the manuscript is well-written but there are some points which can be considered and improved.

- It has not been explained why these two cactus plants was selected.

- The pictures can be shown in more professional and informative way. For example, the figure 1A and B are different in terms of size (one is closer than the other one) and the picture is without scale, same as other figures.

- Figure 3 is not informative in terms of better production of fruits. It is just showing some fruits! There is not any (visual) way to compare that with other treated plants.

- Figure 4 also do not have scale and it is hard to compare the treated plants. It would be great to be replaced with better figures.

- In the result section some abbreviation (experimental groups) is used such as UO. There is no explanation about them except in the legend of the tables or in the method section which is one of the last parts of the text. The abbreviations need to be explained somewhere at the beginning then the reader can grasp it in the first place and does not need to search it.

- There are typos in the text which need to be corrected. E.g. line 308 “ad”

- The material and method is not covering all the methods. E.g. the microbial count and the sugars content or the vitamins. It has to be described or cited with the proper method then it can be evaluated or followed by other researchers.

- The conclusion is just general and vague. It can be more informative.

- The concentration is quite important factor in this work but here it is just briefly pointed out and did not talk more about it in discussion or introduction. It can be reviewed more and even can be explained why the authors selected this concentration in this study. There are recent studies researching about the effect of different concentrations and their growth or suppressing factors for example in case of Ulva on the Arabidopsis.

- There are more recent studies and papers in this field which can be cited.  

- The references should be unified. In some cases, the abbreviation of the journals was used in some cases the whole name of the journals.

Author Response

Responses to the third reviewer

Good evening I am sending you the changes made according to your suggestions.

- It has not been explained why these two cactus plants was selected.

The following sentence was added in the text line 319:

The plants used were chosen because they produce fruit of edible and medicinal interest and could also be appreciated in the future in view of climate change and the low maintenance they require.

- The pictures can be shown in more professional and informative way. For example, the figure 1A and B are different in terms of size (one is closer than the other one) and the picture is without scale, same as other figures.

The Figure 1A and 1B has been improved in size and the view is now more explanatory

- Figure 3 is not informative in terms of better production of fruits. It is just showing some fruits! There is not any (visual) way to compare that with other treated plants.

Figure 3 was removed because it was not scientifically valid. Fruits production has been highlighted in figure 2 together with vegetative growth.

- Figure 4 also do not have scale and it is hard to compare the treated plants. It would be great to be replaced with better figures.

Figure 4 has also been replaced and improved

- In the result section some abbreviation (experimental groups) is used such as UO. There is no explanation about them except in the legend of the tables or in the method section which is one of the last parts of the text. The abbreviations need to be explained somewhere at the beginning then the reader can grasp it in the first place and does not need to search it.

In the results part at the beginning, I have specified each abbreviation so that the reader can immediately understand the meaning. Line 97:

The results show the differences in plant growth, fruit production and sugar and vitamin content obtained with the treatments of (AG) Ecklonia maxima, (HC) Hypnea cornuta, (UO) Ulva ohnoi, and (SM) Sargassum muticum compared with the fertilised control (CTRL).

- There are typos in the text which need to be corrected. E.g. line 308 “ad”

Line 308 has been corrected and the entire text has been checked by an editing agency. I enclose a certificate.

- The material and method is not covering all the methods. E.g. the microbial count and the sugars content or the vitamins. It has to be described or cited with the proper method then it can be evaluated or followed by other researchers.

With regard to the analysis of sugars and vitamins, I have added the relevant reference on the method applied. With regard to microbial analysis, I have added the method and the relevant reference.

Al-Mhanna, N.M.; Huebner, H.; Buchholz, R. Analysis of the Sugar Content in Food Products by Using Gas Chromatography Mass Spectrometry and Enzymatic Methods. Foods. 2018, 7,185.

For microbial count the sentence was added to the text:

Direct determination of the total microbial count by microscopy of the cells contained in a known volume of sample was performed using counting chambers (Thoma chamber). The surface of the slide is etched with a grid of squares, with the area of each square known. Determination of viable microbial load after serial decimal dilutions, spatula seeding (1 ml) and plate counting after incubation.

Prisa, D.; Attanasio, F. Biostimulant derived from the fermentation of Inula viscosa (Inort) in the germination and growth of Amaranthus hypochondriacus. World Journal of Advanced Research and Reviews. 2022, 16, 027–033.

- The conclusion is just general and vague. It can be more informative.

The conclusions have been changed to:

Algae can colonise almost any type of habitat, although most of them live in seas, oceans and fresh waters. However, they can also colonise other environments, including deserts, volcanic waters, very acidic or frozen soils, rocks, plants and artificial substrates. In the terrestrial environment, algae, particularly cyanobacteria, contribute to soil organic matter reserves, either directly or through the secretion of exopolysaccharides and the production of humus-like substances. In addition, they can stimulate the growth of other microflora and microfauna, promote nutrient uptake and participate in organic matter mineralisation processes. Soil degradation, health and environmental risks have been associated with the use of chemical/synthetic fertilisers. Algae serve as an organic fertiliser to increase crop yields. Due to its low cost and environmental friendliness, the use of algae extracts is increasing every day. The results obtained in this experiment provide interesting data on the possible use of new algae as biostimulants for plants, in this case for the cultivation of cacti and succulents, in which there are species that could be interesting from a food point of view.

- The concentration is quite important factor in this work but here it is just briefly pointed out and did not talk more about it in discussion or introduction. It can be reviewed more and even can be explained why the authors selected this concentration in this study. There are recent studies researching about the effect of different concentrations and their growth or suppressing factors for example in case of Ulva on the Arabidopsis.

The concentration and frequency of use in the trial was based on the use of commercial algae, already applied in agriculture. In anticipation of the possible use of these algae and their possible marketing, an attempt was made to compare the protocols and concentrations of use with already tried and tested products.

In any case, I have included some references of tests carried out with algae on other plant species with the relative concentration of use.

Prisa, D. Ascophyllum nodosum extract on growth plants in Rebutia heliosa and Sulcorebutia canigueralli. GSC Biological and Pharmaceutical Sciences. 2020, 1, 39–45.

Prisa, D.; Gobbino, M. (2021). Microbic and Algae biofertilizers in Aloe barbadensis Miller, Open Access Research Journal of Biology and Pharmacy. 2021, 1, 1-9.

Prisa, D. Biological mixture of brown algae extracts influences the microbial community of Lobivia arachnacantha, Lobivia aurea, Lobivia jajoiana and Lobivia Grandiflora in pot cultivation. GSC Advanced Research and Reviews. 2021, 3, 43–53.

- There are more recent studies and papers in this field which can be cited. 

I have included more recent bibliography:

Campobenedetto, C.; Agliassa, C.; Mannino, G.; Vigliante, I.; Contartese, V.; Secchi, F.; Bertea, C.M. A biostimulant based on seaweed (Ascophyllum nodosum and Laminaria digitata) and yeast extracts mitigates water stress effects on tomato (Solanum lycopersicum L.). Agriculture. 2021, 11, 557–572.

Castellanos-Barriga, L.G.; Santacruz-Ruvalcaba, F.; Hernández-Carmona, G.; Ramírez-Briones, E.; Hernández-Herrera, R.M. Effect of seaweed liquid extracts from Ulva lactuca on seedling growth of mung bean (Vigna radiata). Journal of  Applied Phycology. 2017, 29, 2479–2488.

De Saeger, J.; Van Praet, S.; Han, T.; Depuydt, S. Toward the molecular understanding of the action mechanism of Ascophyllum nodosum extracts on plants. Journal of  Applied Phycology. 2020, 32, 573–597.

Gupta, S.; Stirk, W.A.; Plačková, L.; Kulkarni, M.G.; Doležal, K.; Van Staden, J. Interactive effects of plant growth-promoting rhizobacteria and a seaweed extract on the growth and physiology of Allium cepa L. (onion). Journal of Plant Physiology. 2021, 262, 153437.

Hussain, H.I.; Kasinadhuni, N.; Arioli, T. The effect of seaweed extract on tomato plant growth, productivity and soil. Journal of Applied Phycology. 2021, 33, 1305–1314.

Kocira, S.; Szparaga, A.; Kuboń, M.; Czerwińska, E.; Piskier, T. Morphological and biochemical responses of Glycine max (L.) Merr. To the use of seaweed extract. Agronomy. 2019, 9, 93.

Kulkarni, M.G.; Rengasamy, K.R.R.; Pendota, S.C.; Gruz, J.; Plačková, L.; Novák, O.; Doležal, K.; Van Staden, J. Bioactive molecules derived from smoke and seaweed Ecklonia maxima showing phytohormone-like activity in Spinacia oleracea L. N Biotechnology. 2019, 48, 83–89.

Mahmoud, S.H.; Salama, D.M.; El-Tanahy, A.M.M.; Abd El-Samad, E.H. Utilization of seaweed (Sargassum vulgare) extract to enhance growth, yield and nutritional quality of red radish plants. Annals of Agriculture Science. 2019, 64, 167–175.

Shukla, P.S.; Shotton, K.; Norman, E.; Neily, W.; Critchley, A.T.; Prithiviraj, B. Seaweed extract improve drought tolerance of soybean by regulating stress-response genes. AoB Plants. 2018, 10, 1–8.

- The references should be unified. In some cases, the abbreviation of the journals was used in some cases the whole name of the journals.

I standardised the references by inserting the whole name:

Sarwar, M.; Saleem, M.F.; Ullah, N. Nutrition in Alleviation of Heat Stress in Cotton Plants Grown in Glasshouse and Field Conditions. Scientific Reports. 2019, 13022, 277.

Stirk, W.A.; Van Staden, J. Plant Growth Regulators in Seaweeds. In Advances in Botanical Research Elsevier. 2014, 71, 125–159.

Moheimani, N.R.; BorowitzKa, M.A. The long term culture of the coccolithophore Pleurochrysis carterae (Haptophyta in outdoor raceway ponds. Journal of Applied Phycology. 2006, 18, 703-712.

Pushparaj, B.; Pelosi, E.; Tredici, M.R.; Pinzani, E.; Materassi, R. (1997). An integrated culture system for outdoor production of microalgae and cyanobacteria. Journal of Applied Phycology 1997,9, 113-119.

El Arroussi, H.; El Mernissi, N.; Benhima, R. Microalgae polysaccharides a promising plant growth biostimulant. Journal Algal Biomass Utln. 2016, 7,55–63.

Garcia- Gonzalez, J.; Sommerfeld, M. Biofertiliser and biostimulant properties of the microalga Acutodesmus dimorphus., Journal of Applied Phycology 2016, 28, 1051–1061.

Grima, E.M.; Belarbi, E.H.; Fernandez, F.A. Recovery of microalgal biomass and metabolites: Process options and economics. Biotechnology Advances, 2003, 20, 491–515.

Kumar, K.S.; Dahms, H.U.; Won, E.J. Microalgae. A promising tool for heavy metal remediation. Ecotoxicology and Environmental Safety 2015, 113, 329–352.

Kumar, M.; Prasanna, R.; Bidyarani, N.  Evaluating the plant growth promoting ability of thermotolerant bacteria and cyanobacteria and their interactions with seed spice crops. Scientia Horticulturae. 2013, 17, 94–101.

Mulberry, W.; Konrad, S.; Pisarro, C. Bio fertilizers from algal treatment of dairy and swine manure effluents, Journal of Vegetable Science. 2007, 12, 107–125.

Pushparaj, B.; Pelosi, E.; Tredici, M.R. An integrated culture system for outdoor production of microalgae and cyanobacteria. Journal of Applied Phycology. 1997, 9, 113–119.

Hernandez-Carlos, B.; Gamboa-Angulo, M.M. Metabolites from freshwater aquatic microalgae and fungi as potential natural pesticides. Phytochemestry Reviews. 2011, 10, 261–286.

Faheed, F.A.; El Fattah, Z.A. Effect of Chlorella vulgaris as bio-fertilizer on growth parameters and metabolic aspects of lettuce plant. Journal of Agriculture and Social Sci-ence. 2008,4,165-169.

Chiaiese, P.; Corrado, G.; Colla, G.; Kyriacou, M.C.; Rouphael, Y. Renewable sources of plant biostimulation: Microalgae as a sustainable means to improve crop preformance Frontier in Plant Science. 2018, 9, 1782.

Spolaore, P.; Joannis-Cassan, C.; Duran, E.; Ismbert, A. Commercial applications of microalgae. Journal of Bioscience and Bioengineering. 2006, 101, 87-96.

Tibbetts, S.M.; Milley, J.E.; Lall, S.P. (2015). Chemical composition and nutritional properties of fresh water and marine microalgal biomass cultured in photobioreactors. Journal of Applied Phycology. 2015, 27,1109-1119.

The entire text was edited with a specialised agency, whose certificate I enclose.

I would like to thank you for your suggestions and help in improving my scientific article.

Reviewer 2 Report

It is recommended to search for more recent citations related to the topic and to improve the introduction.

A broader discussion of the results is recommended.

In materials and methods, specify more about the methodology and characteristics of the equipment used.

In conclusions specify more about the results.

Author Response

Good morning I wanted to thank you for your review work. I have edited and added all the parts you mentioned. And I have increased the number of references.

The text has been edited by a professional agency. Thank you

Line 44. Indicate in which plants the plant extracts have been previously used.

Previous experiments using algae on plants such as maize and rice resulted in an increase in growth in terms of dry matter and the C and N content of the soil. In addition, an increase in shoot height in the treated maize theses and an increase in root and shoot development in rice were observed. Tests on Amaranthus spp. showed an increase in plant height and biomass produced in the theses treated with algae. Further trials on Portulaca grandiflora, Aloe Barbadensis Miller and Lobivia spp. showed a significant effect on increased vegetative and root development of the plants and on the mineral and sugar content in stems and leaves.

D’Acqui LP, et al. (2006). Use of indigenous N2-fixing cyanobacteria for sustainable improvment of soil biogeochemical performance and physical fertility in semiarid tropics. Final Report. EU, Brussels: EU-ICA4-CT-2001-10058. Pp. 1-300

D’Acqui L.P., Maliondo S.M., Malam Issa O., Le Bissonnais Y., Ristori G.G. (2004). Influence of indigenous strains of cyanobacteria on physical and biochemical properties of tropical soils. In: Abstracts of the 4th International Symposium of the Working group MO-IsMOM 2004. Wuhan, p.110

D’Acqui L.P. (2016). Use of Indigenous Cyanobacteria for Sustainable Improvement of Biogeochemical and Physical Fertility of Marginal Soils in Semiarid Tropics. In: Eds. N.K. Arora, S. Mehnaz, R. Balestrini (Eds), Bioformulations: for Sustainable Agriculture, Springer. Doi 10.1007/978-81-322-2779-3

  1. Prisa, (2019). Possible use of Spirulina and Klamath algae as biostimulant in Portulaca grandiflora (Moss Rose). World Journal of Advanced Research and Reviews, 2019, 3(2), 1-6.

D.Prisa, M. Gobbino (2021). Microbic and Algae biofertilizers in Aloe barbadensis Miller, Open Access Research Journal of Biology and Pharmacy, 1(2): 1-9

D.Prisa (2021). Biological mixture of brown algae extracts influences the microbial community of Lobivia arachnacantha, Lobivia aurea, Lobivia jajoiana and Lobivia grandiflora in pot cultivation. GSC Advanced Research and Reviews, 2021, 08(03), 043–053

Line 60. Indicate which are the biofertilizers that have been generated based on extracts of different species of algae.

From the different algal species, biofertilisers with stimulating properties are generally produced individually or in combination with bacteria and mycorrhizae. Microalgae and cyanobacteria products are generally developed by means of photobioreactors, such as: Arthrospira spp., Dunaliella spp., Anabaena spp., Phaeodactylum spp., Pleurochrysis spp., Chlorella spp. and Nannochloropsis. These products once obtained can be applied to plants in fertigation or combined with fertilisers in integrated agriculture protoclimates. Stimulant products based on Ascophyllum nodosum are used today as stimulants and for increasing plant protection from biotic and abiotic stresses in ornamentals.

Moheimani N.R., BorowitzKa M.A. (2006). The long term culture of the coccolithophore Pleurochrysis carterae (Haptophyta in outdoor raceway ponds. J. Appl.Phycol. 18, 703-712

Pushparaj B., Pelosi E., Tredici M.R., Pinzani E., Materassi R. (1997). An integrated culture system for outdoor production of microalgae and cyanobacteria. J. Appl. Phycol. 9, 113-119

Radmann E.M., Rheinehr C.O., Costa J.A.V. (2007). Optimisation of the repeated batch cultivation of microalga Spirulina platensis in open raceway ponds. Aquaculture, 265, 118-126

  1. Prisa, (2020). Ascophyllum nodosum extract on growth plants in Rebutia heliosa and Sulcorebutia canigueralli. GSC Biological and Pharmaceutical Sciences, 2020, 10(01), 039–045

Line 269. Mention how climate change can affect crops positively and negatively.

Climate change affects the shape of plants and their fruit, and in flower development and colour intensity. The effect on plants is well evident, even when they are cultivated and well watered, becoming smaller, shorter and less drought-resistant. This also affects the reduction in productivity on the amount of CO2 they absorb for photosynthesis, which drops significantly. Climate change can also favour the development of serious plant diseases due to fungi or insects that normally live in different areas.

Bhadra P., Maitra S., Shankar T., Hossain A., Praharaj S.,  Aftab T. (2022). Climate change impact on plants: Plant responses and adaptations. Plant Perspectives to Global Climate Changes, Academic Press. Pages 1-24.

Line 283 and 284. Explain the positive effects on plant growth and yield of the application of algae extracts and that are not attributed to the supply of nutrients.

Studies by Faheed and El Fattah (2008) who studied the effects of the algae Chlorella vulgaris on Lactuca sativa found that the presence of the algae accelerated the germination process of the seeds and increased the chlorophyll a and b and carotenoid content. Agwa et al. (2017) using C. vulgaris on Hibiscus esculentus replacing chemical fertilisers showed improved plant growth and biomass production. Algae have a high content of primary metabolites, carbohydrates and lipids (55-70% fresh weight). Generally, in Chlorella spp., Chlamydomonas spp., Dunaliella spp. and Spirulina spp., carbohydrates can be up to 46% of the dry extract, while protein accounts for 18-46% of the dry extract. The presence of certain amino acids, such as tryptophan and arginine in algae extracts, causes a significant increase in growth and yield of the cultures, because these amino acids are the metabolic precursors of phytohormones. Algae also exhibit an indirect biostimulating effect, attributed to their ability to increase and modify the microbial component of the soil, also with concomitant effects on nutrient mineralisation. Several studies have demonstrated the succession of the soil microbiota following the inoculation of algae, whose prolific production of extracellular polymeric substances served as a carbon source for plants and bacteria in the rhizosphere.

Agwa O.K., Ogugbue C.J., Williams E.E. (2017). Field evidence of Chlorella vulgaris potentials as a biofertilizer for Hibiscus esculentus. Int J Agric Res., 12: 181-189

Chiaiese P., Corrado G., Colla G., Kyriacou M.C., Rouphael Y. (2018). Renewable sources of plant biostimulation: Microalgae as a sustainable means to improve crop preformance Front. Plant. Sci. 9, 1782

Faheed F.A., El Fattah Z.A. (2008). Effect of Chlorella vulgaris as bio-fertilizer on growth parameters and metabolic aspects of lettuce plant. J Agric Social Sci 4:165-169

Spolaore P., Joannis-Cassan C., Duran E., Ismbert A. (2006). Commercial applications of microalgae. J.Biosc. Bioeng. 101, 87-96

Tibbetts S.M., Milley J.E., Lall S.P. (2015). Chemical composition and nutritional properties of fresh water and marine microalgal biomass cultured in photobioreactors. J.Appl.Phycol. 27,1109-1119

Line 291. Mention which are the nutraceutical parameters of the fruits that are increased by the application of algae extracts.

Treatments with algae such as Macrocystis pyrifera, Bryothamnion triquetrum, Ascophyllum nodosum, Grammatophora spp., Macrocystis integrifolia on Cucumber (Cucumis sativus L.) Fruit showed a better yield than the control and an increase in antioxidant capacity and vitamin C content. Algae treatments on Annurca apple fruit at harvest showed an increase in total soluble solids content (TSS), total acidity (TA), pH, flesh firmness and red colouration of the epicarp. The treated fruits also showed, during cold storage, a significantly higher content of total polyphenols in the pulp and a higher concentration of xyloglucoside and floridzin and an increase in flavonols.

Trejo Valencia, R.; Sánchez Acosta, L.; Fortis Hernández, M.; Preciado Rangel, P.; Gallegos Robles, M.Á.; Antonio Cruz, R.d.C.; Vázquez Vázquez, C. Effect of Seaweed Aqueous Extracts and Compost on Vegetative Growth, Yield, and Nutraceutical Quality of Cucumber (Cucumis sativus L.) Fruit. Agronomy 20188, 264.

Graziani, G.; Ritieni, A.; Cirillo, A.; Cice, D.; Di Vaio, C. Effects of Biostimulants on Annurca Fruit Quality and Potential Nutraceutical Compounds at Harvest and during Storage. Plants 20209, 775.

Line 340 and 341. Include the citation of the methodology to determine the content of sugars, vitamin C, vitamin A and vitamin E.

Al-Mhanna N.M., Huebner H., Buchholz  R. (2018). Analysis of the Sugar Content in Food Products by Using Gas Chromatography Mass Spectrometry and Enzymatic Methods. Foods.  7(11): 185.

Reviewer 3 Report

The research aimed to test different seaweed extracts, derived from three macroalgae representative of the Rhodophyta, Chlorophyta, and Phaeophyceae, as a biofertilizer for the growth of Mammillaria prolifera and Mammillaria glassii and the production of edible fruits, and to evaluate the interactions between algae and the soil microbiome.

In my opinion the work was not well written. For example, lines 94-95: the sentence in uncomplete. Line 98: there is a repetition of “in” “in”.

Before to use the acronyms UO, SM, AG,… the authors have to explain the meaning of this acronyms in the text.

Line 113: you can not repeat “in” “in”.

There is not a real discussion of results.

In my opinion, the work have to be rewritten, so it can not accepted in the current form.

Best regards

Author Response

Good evening I am sending you the changes made according to your suggestions.

I have modified and expanded the sentence in lines 94-95 by better describing the effects found on both plant growth and the production of fruit and nutraceutical compounds. I corrected line 98 and 13 and generally reformulated all the results. I specified the meaning of the acronyms referring to the algae UO, SM, AG. The reference was already present in Materials and Methods, but according to your instructions I have also included it in the results. In materials and methods I have included a citation concerning the method of analysis for sugars and vitamins.

In agreement with the first reviewer, the introductory part was improved and expanded with new references. The discussions have also been reworded as requested and new bibliographical references have been inserted.

The entire text was edited with a specialised agency, whose certificate I enclose.

Thank you
